# The Effect of the Periodic Drying Method on the Drying Time of Hazelnuts and Energy Utilization

**DOI:** 10.3390/foods13060901

**Published:** 2024-03-15

**Authors:** Mithat Akgün, Emrullah Kontaş

**Affiliations:** Department of Renewable Energy, Institute of Science and Technology, Ordu University, Ordu 52200, Türkiye; emrullahknts@gmail.com

**Keywords:** *Corylus avellana*, energy saving, hazelnut food properties, mass loss

## Abstract

Hazelnut is a shelled fruit that is stored by drying and used as a snack or in industry. Since the hazelnut drying process is energy-intensive, there is a need for drying methods that will reduce the energy cost without lengthening the drying time. In this study, the effects of periodic drying of hazelnuts’ energy recovery, oil, and protein content, as well as mass losses, were studied. Fresh Tombul hazelnuts (*Corylus avellana* L.) with a diameter of Ø 15–16 mm were dried in a tunnel dryer over 16 different periods by adjusting the drying time inside and waiting time outside the oven until the moisture content reached 6%. Drying experiments were carried out at 45 °C and three different air velocities. The increase in air velocity resulted in a reduction in the periodic drying time between 10% and 36%. The optimum drying in terms of drying time and energy utilization was realized at 0.5 m/s air velocity, with a 1.5 h working time and 0.5 h waiting time. During this period, drying time increased by 19% and energy utilization was 69%. For periodic drying, the increase in oven working time causes a decrease in energy utilization, while the increase in waiting time causes an increase in energy utilization and drying time. Periodic drying had no negative effect on hazelnut oil and protein content. Periodic drying is a suitable option for saving energy during hazelnut drying.

## 1. Introduction

In the Black Sea coastal region of Turkey, although there various types of hazelnuts are commonly grown, Tombul hazelnut (*Corylus avellana* L.) is the most preferred type for oil production and snacks due to its high oil content (69–72%), distinctive flavor, and complete husk removal (blanching) [1,2]. Hazelnut is a fruit that is preserved by drying in the shell. Depending on the harvest time, altitude, and variety, hazelnuts’ initial moisture content ranges from 18% to 38%. The drying process is essential to reduce this moisture content to an equilibrium moisture of 6% or below. The process of reducing the moisture value of hazelnuts by drying can be achieved by different methods depending on the amount of product, climatic conditions during the harvest period, and altitude [3]. Hazelnuts are traditionally dried in the sun by laying on the ground (on the soil). With climate change, sun-drying hazelnuts has become either impractical or time-consuming. Drying hazelnuts on the ground exposes them to quality losses, aflatoxin formation, and increased costs [4,5,6]. Therefore, the mechanization of hazelnut drying has become increasingly imperative. However, attention should be paid to the selection of a method that will reduce the costs associated with mechanization and minimize quality losses in hazelnuts [7,8,9,10]. The protein and oil content of hazelnuts, which are particularly important for human health in hazelnuts, have been reported to vary according to the drying method [11]. There a number of studies in the literature have deal with the composition of the Tombul hazelnut [12,13].

The drying process is an energy-intensive operation, constituting 90% of the energy costs for food-processing facilities. Drying machines typically operate at 25–50% thermal efficiency, reflecting their relatively low energy utilization. In some countries, energy expenditure in the drying industry accounts for about 7–15 per cent of the total energy consumption of the industry [14,15]. One of the most important energy losses caused by convection food drying systems is the exhausted moist hot air. Exhausted energy accounts for about 35–45% of the total energy consumption in drying and contributes to the high greenhouse gas emissions [16,17].

Numerous scientific drying studies are being conducted to preserve food characteristics while reducing the drying time and lowering the energy costs [8]. During the selection of methods for these drying studies, new perspectives have emerged, considering not only economic factors, time, and the preservation of food properties but also the conservation of the characteristic features of the food, the expected qualities of the dried product (such as color, aroma, appearance, size, shape, and hygiene), the quantity that is to be dried, and the environmental impacts. These novel methods include microwave-drying, heat-pump-drying, infrared-drying, freeze-drying, vacuum-drying, LED-drying, and hybrid drying systems. The reduction in drying time is inversely proportional to the increase in microwave power [9,18]. The low energy consumption of heat-pump-drying systems contributes to the increase in the products’ price competitiveness [19,20]. Infrared drying technology offers advantages such as a high energy efficiency, a short drying time, an even heating of materials, easy control of material temperature, high-quality final products, and low energy costs [21]. Freeze-drying leads to relatively high nutrient retention, size stabilization, color retention and a good rehydration ability, but has a high drying cost [22]. It was revealed that the drying time of fruits in the drying process with light emitting diodes (LEDs) was shorter than convective dryers and the drying time decreased as the LED color temperature increased [23,24]. In addition to these methods, hybrid drying studies have been carried out recently, combining drying methods to make food drying faster and more economical [25,26,27,28]. 

During the convective drying of hazelnuts, the mass transfer of moisture from the hazelnut kernel to the shell and then from the shell to the air is shown schematically in Figure 1. In the drying process, the shell initially dries first, but the kernel hazelnut preserves its moisture. Throughout the drying time, the moisture difference between the kernel hazelnut and the shell decreases, the shell hardens, and the gap between the hazelnut and the shell increases, making the mass transfer from the kernel hazelnut to the air even more challenging over time. Meanwhile, conditioned air continuously flows over the shell; however, as there is no moisture in the shell, the mass transfer from the shell to the air decreases. The drying system continues to operate, expending energy unnecessarily for heating and moving the air. This drying issue is encountered in the drying of all nuts with shells, such as hazelnuts. The solution to this problem is to wait for the moisture to reach the shell from the internal (kernel) hazelnut (waiting time) and then dry the product before its utilization (working time). This repeated process of drying the product is called the “M Periodic Drying Method”. Traditional drying methods are not preferred by hazelnut producers and industrialists due to their high energy costs. Therefore, there is a need to develop a more economical hazelnut drying system. In addition, there is no study in the literature on the periodic drying of hazelnuts. This experimental study was carried out to fill a gap in the literature and to determine the effects of drying on energy recovery, hazelnut oil and protein content, and mass loss. 

## 2. Materials and Methods

### 2.1. Materials and Drying Experiments

In this study, Tombul hazelnut (*Corylus avellana* L.) was used. Hazelnut husk (a), shelled and unshelled fresh hazelnut (b), and dried shelled and unshelled hazelnut (c) pictures are shown in Figure 2. Hazelnut was harvested fresh from Ordu province, which is the largest hazelnut producer in Turkey. Fresh hazelnuts were harvested from the branch with husk, and after the hazelnut was removed from the husk, it was kept in a cooler at +4 °C. The hazelnuts were taken from the cooler and dried in a tunnel dryer. Drying experiments were carried out at a temperature of 45 °C and air velocities of 0.5 m/s, 1 m/s, and 1.5 m/s. Although Tombul hazelnut fruits are in the size range of Ø 13–19 mm, shelled hazelnuts of Ø 15–16 mm were used in this study.

When hazelnuts were harvested, their moisture content was approximately 28%. Through the drying process, the equilibrium moisture content was reduced to 6%. Each drying experiment utilized 100 g of fresh hazelnuts. 

In this study, despite hazelnuts being periodically dried, continuous drying and sun-drying were also conducted for a comparison of the time taken and comparative food analyses. A schematic view of the tunnel-type conventional dryer used for drying is given in Figure 3. During sun-drying, the speed of air passing over hazelnuts was measured with a thermo-anemometer, the ambient temperature was measured with a K-type thermo-couple, and the hazelnut surface temperature was measured with an infrared temperature gauge. During periodic drying, the air velocity passing over the hazelnuts in the tunnel dryer was measured with a hot wire anemometer (telescopic probe) and the air temperature was measured with a K-type thermocouple. The fan speed on the control panel was mechanically adjusted to set the air speed, while the temperature was set to 45 °C. Hazelnut mass losses were measured with an electronic balance with a sensitivity of 0.01 g. Hazelnut moisture content was determined using an infrared moisture meter, Precisa XM60 (Precisa Gravimetrics AG, Dietikon, Switzerland).

As given in Table 1, hazelnut drying experiments were carried out in 16 periods depending on the periodic drying, the time spent drying in the oven (working time), and the time spent idle outside the oven (waiting time). Table 1 shows the drying process of hazelnuts. Case 1 refers to the hazelnuts that were dried in the oven for 0.5 h (0.5 h work) and then left outside the oven for 0.5 h (0.5 h wait) until they were completely dry. After removing the hazelnut sample container from the oven, it was covered with a lid to protect it from external influences and kept at room temperature (approximately 30 °C) during the waiting period.

### 2.2. Energy Utilization

The energy utilization of periodic drying compared to continuously operating conventional convection dryers is obtained from the following equations.

The energy flow rate absorbed by the air used for drying [29,30]:(1)$$E=\dot{m}c_{p}\Delta T$$

Mass flow rate of air [29,30]:(2)$$\dot{m}=\rho vA$$

The second equation is substituted into the first equation [29,30]:(3)$$E=\rho vAc_{p}\Delta T$$

For ${\;T}_{a}$ = 37.3 °C, *ρ* = 1.1397 kg/m^3^, *C_p_* = 1.0063 kJ/kg °C, $\Delta T=T_{o}-T_{e}$ = 45 − 30 = 15 °C, *A* = 0.25 m^2^.

Total energy flow rate consumed during continuous and periodic drying [30]:(4)$$E_{c}=E_{a}t_{c}$$
(5)$$E_{p}=E_{a}t_{p}$$

Specific energy consumption (SEC) for the evaporation of water from hazelnuts [31,32]:(6)$$SEC=\frac{E_{d}}{m_{w}}$$

The energy utilization of periodic drying compared to continuous drying [33]:(7)$${\%ɳ}_{e}=100-\left(E_{p}/E_{c}\right)100$$

The moisture content of the samples was calculated using Equation (8) [34]:(8)$$M=\frac{{m_{t}-m}_{dm\;}}{m_{dm}}$$

### 2.3. Food Analysis

In the study, protein content was determined according to Venkatachalam and Sathe [35] and oil content was determined according to Firestone [36].

### 2.4. Statistical Analysis

The changes in hazelnuts treated with different drying methods were analyzed with XLSTAT version 2020 (Microsoft Excel). Figures were prepared in Excel and Sigma Plot (14 version). Two-way ANOVA and Türkiye multiple comparison test were used.

## 3. Results and Discussion

The outcomes of this study were classified and discussed under different headings as drying experiments, energy utilization, and protein and oil analyses in hazelnuts.

### 3.1. Drying Experiments

Drying Experiments: In this experimental study, time-dependent moisture content was obtained for each periodic drying. It is not feasible to present all the results graphically. Thus, in this article, the effects of drying air velocity during periodic drying are presented via graphs for each operating period for comparison. The other figures are given as Appendix A. The time-dependent moisture contents for three different air velocities and each operating period are shown in Figure 4, Figure 5, Figure 6 and Figure 7.

As seen in Figure 4, the drying time under case 1 conditions was 1290 min at air velocities of 1.5 m/s and 1 m/s, while this duration increased by 9.6% at an air velocity of 0.5 m/s. It is evident from Figure 5, Figure 6 and Figure 7 that as the air velocity decreases in case 6, case 11, and case 16, the hazelnut drying time increases by approximately 10%. As the case number increases, the drying time also increased by an average of 5%.

As observed in Figure 5, Figure 6 and Figure 7, as the air velocity increases, the total drying time shortened between 10% and 36%. As the case number increases, when transitioning from an air velocity of 1 m/s to 1.5 m/s, the drying time decreases by 9–14%, while when transitioning from an air velocity of 0.5 m/s to 1 m/s, the drying time decreases by 7–14%. This decrease occurred approximately proportionally with the increase in the case number. The reduction in hazelnuts’ drying time due to the increase in air velocity during periodic drying is similar to the situation of continuous drying described in the literature [12].

Although the drying process was conducted at three air velocities in the study, comparison graphs for working periods are provided for the air velocity of 1.5 m/s, which has the shortest drying times, in Figure 8, Figure 9, Figure 10 and Figure 11.

As depicted in Figure 8, it can be observed that, compared to continuous drying, as the idle time increases in the 0.5 h working periods, the total drying time also increases. The increases in drying time compared to continuous drying for cases 1, 2, 3, and 4 are 19%, 44%, 58%, and 83%, respectively.

As seen in Figure 9, it can be observed that, similar to Figure 8, as the waiting time increases in the 1 h working periods compared to continuous drying, and the total drying time also increases. The increases in drying time compared to continuous drying for cases 5, 6, 7, and 8 are 22%, 33%, 55.5%, and 74%, respectively.

As the waiting time increases in the 1.5 h working periods, the total drying time for the 1.5 m/s air velocity also increased compared to continuous drying, as shown in Figure 10. This time increase was 8%, 19%, 25%, and 42% for cases 9, 10, 11, and 12, respectively.

Compared to continuous drying, the total drying time increased as the idle time outside the oven increased in the 2 h operation period. This increase was 6%, 8%, 22%, and 22% for case 13, 14, 15, and 16, respectively (Figure 11).

During periodic drying, as the oven working time increases, the hazelnuts’ drying time decreases, and as the waiting time outside the oven increases, the drying time also increases. Compared to continuous drying (1080 min), for a 1.5 m/s air velocity, the shortest drying time (1140 min) occurs in case 13, while the longest drying time (1980 min) occurs in case 4. Since there are no data on periodic drying in the literature, a comparison with the results obtained in this study could not be made.

The time-dependent mass losses during sun-drying and the temperature changes that occurred during the drying process are given in Figure 12.

As observed from Figure 12, hazelnuts dried in the sun over 4 days. The daytime ambient temperature (35.2 °C) and hazelnut surface temperature reached a maximum (53.7 °C) between 13:00 and 14:30. Due to the high humidity (highest at 93% and lowest at 75%) in the region, hazelnuts absorb moisture during the night. Hazelnuts can only release this moisture after the sun rises, typically within 2–3 h. The time-dependent mass loss curves in solar-drying and the emphasis on the difference between the hazelnuts’ top temperature and ambient temperature are consistent with the studies of Kandemir [12]. The drying time of hazelnuts under sunlight is similar to the studies of Islam and Turan [37].

The working time of the oven and the idle waiting time outside the oven of the hazelnuts according to the air velocities are given in Table 2. Here, only the last times are given for each period.

As seen in Table 2, the shortest operating time and the longest total drying time occurred during case 4 at an air velocity of 1.5 m/s. In all experiments, the shortest drying times compared to continuous drying were observed in the 0.5 h waiting periods.

The shortest working time and the longest waiting time for all three air velocities are observed in the 0.5 h working and 2 h waiting periods.

### 3.2. Energy Utilization

The results of this study included the M. Periodic Drying Model, aiming to reduce the drying energy costs in conventional drying systems and, consequently, reduce the carbon footprint during drying. These were examined in terms of their energy consumption and energy utilization. The specific energy consumption of the channel-type dryers used for drying hazelnuts is given in Figure 13, Figure 14 and Figure 15 for continuous drying and periodic drying.

When the graphs in Figure 13, Figure 14 and Figure 15 are examined, it can be seen that the lowest specific energy consumption per kg water mass that evaporated from hazelnut was obtained during the 0.5 h work and 2 h wait periods. In hazelnut drying, the lowest specific energy consumption per kg water mass was obtained as 645.1 kWh/kg water in Case 4 for 0.5 m/s air velocity, while the highest energy consumption was obtained as 3096.8 kWh/kg water in Case 16 with 1.5 m/s air velocity. The specific energy consumptions of continuous drying, depending on the air velocities (0.5 m/s, 1 m/s, and 1.5 m/s), were calculated as 1600.3 kWh/kg water, 3354.6 kWh/kg water and 4644.8 kWh/kg water, respectively. As seen in the graphs, the specific energy consumption for drying increases proportionally with the increase in air velocity. The specific energy consumption values obtained during continuous drying are similar to those of Motevali et al. [31] and Nwakuba [32]. However, there are no data in the literature related to the specific energy consumption values of periodic drying. Therefore, the results of periodic drying could not be compared with the literature findings. The specific energy consumption of periodic drying is lower than that of continuous drying for each period.

The energy utilization calculated from Equation (7) for periodic drying compared to the continuous drying at three different air velocities is given in Table 3, Table 4 and Table 5.

Table 3 indicates that, at an air velocity of 0.5 m/s, the highest energy of periodic drying compared to continuous drying is 59.7%, obtained in case 4, while the lowest energy utilization is 3.3%, obtained during periods 13 and 14. At an air velocity of 1 m/s, the energy utilization compared to continuous drying is 61.5% in case 4, while the lowest energy utilization was calculated as 17.9% during periods 13 and 14 (Table 4). Similarly, at a low air velocity of 1.5 m/s, the energy utilization compared to continuous drying is 61.1%, as obtained in case 4, with the lowest energy utilization determined to be 11.1% in case 13 (Table 5).

It is evident from the energy utilization tables that, as the air velocity increases, the energy utilization from periodic drying compared to continuous drying increases. However, as the oven operation time increases, the energy utilization decreases, and while the energy utilization increases, the idle waiting time outside the oven increases.

### 3.3. Protein and Oil Analyses in Hazelnut

The hazelnut oil and protein percentages under different drying conditions are presented in Table 6. As seen in Table 6, the lowest oil percentage occurred under the drying conditions of case 10, at 61.46%. The highest oil percentage, 67.71%, was observed under the drying conditions of Period 5, with a 10.33% increase in oil percentage. The oil percentages obtained during sun-drying and continuous drying are similar to the other periodic drying conditions.

When compared to continuous drying (control sample) for all periodic drying conditions, the lowest protein percentage was obtained under the drying conditions of case 16 (14.8%), while the highest protein percentage was measured under the drying conditions of case 7 (17.8%). However, the protein values for sun-drying and periodic drying are similar.

Protein and oil content may vary according to the drying method, the variety, and the size of the hazelnut [12,37,38]. In this study, in order to minimize variability, hazelnuts were harvested from a single orchard, hazelnut size was kept constant, and a single variety was used as the material. The results of the present study related to the oil and protein content of the hazelnuts were in good agreement with those of the literature. The values measured in all methods were between the values specified by Balık et al. [13] and Sali [38].

## 4. Conclusions

This experimental study showed that increasing the air velocity resulted in a decrease in total drying time, with a range of 10–36%. Additionally, as the number of cases increased, the total drying time decreased by 9–14%, from 1 m/s to 1.5 m/s, and by 7–14%, from 0.5 m/s to 1 m/s. Although the moisture content–time curves for periodic drying were very similar to those for continuous drying, the increase in waiting time during the operating periods resulted in an average increase of 20% in periodic drying times compared to continuous drying. The increase in drying time ranged from 19% to 83% from case 1 to case 4, respectively. The drying times were shortest during the 0.5 h wait periods, while the shortest oven working times were observed during the 0.5 h work periods. Periodic drying resulted in an increase in oven working time and a decrease in specific energy consumption, while the increase in waiting time led to a decrease in specific energy consumption to continuous drying. The minimum specific energy consumption (645.1 kWh/kg water) was realized for the 0.5 m/s air velocity at a 0.5 h work and 2 h wait period, while the maximum specific energy consumption (3096.8 kWh/kg water) was realized for a 1.5 m/s air velocity at a 2 h work and 0.5 h wait period.

Case 4 had the highest energy utilization (74.1%) at 1.5 m/s air velocity, while cases 13 and 14 had the lowest (10%) at 0.5 m/s air velocity. The optimal drying occurred at an air velocity of 0.5 m/s with a 1.5 h work and 0.5 h wait period, considering both drying time and energy utilization. During this period, the drying time increased by 19%, and the energy utilization was 69%. The effects of periodic drying on the protein and oil content of hazelnuts were similar to those of the conventional and sun-drying methods. In conclusion, the ‘M Periodic Drying Method’ is considered a viable option for reducing energy costs in hazelnut drying.

## Figures and Tables

**Figure 1 foods-13-00901-f001:**
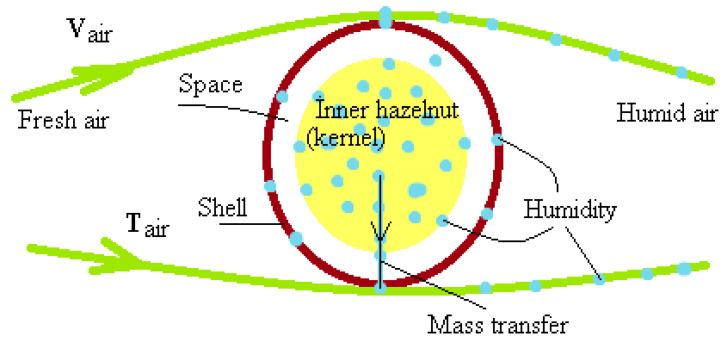
Schematic representation of the mass transfer from the inner hazelnut to the air.

**Figure 2 foods-13-00901-f002:**
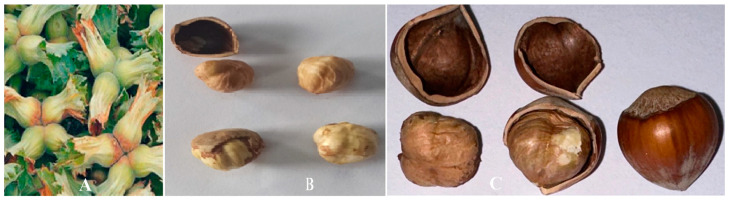
Hazelnut husk (**A**), shelled and kernel fresh hazelnut (**B**), and dried shelled and kernel hazelnut (**C**) pictures.

**Figure 3 foods-13-00901-f003:**
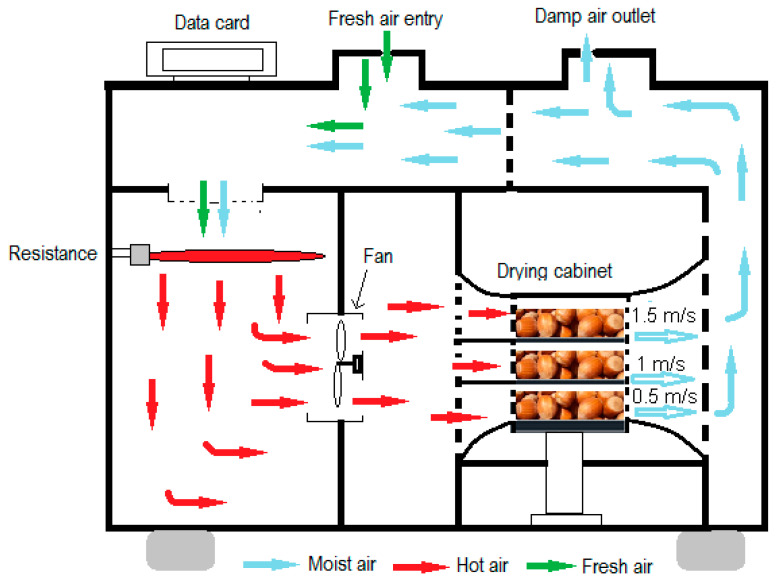
Schematic view of convective tunnel-type dryer.

**Figure 4 foods-13-00901-f004:**
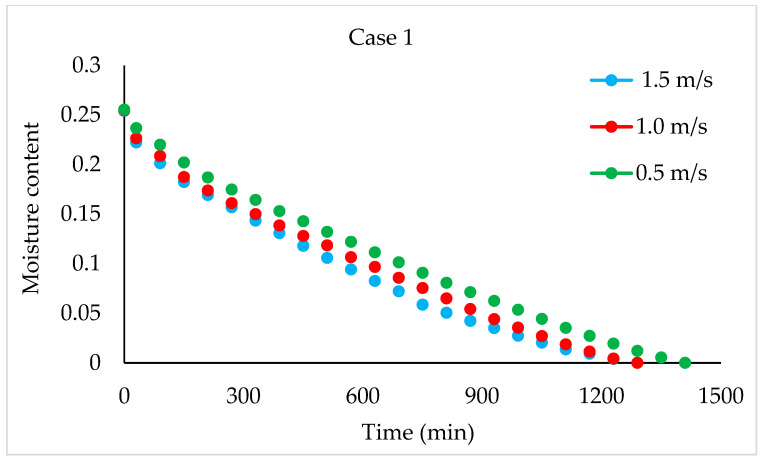
Changes in time-dependent moisture content with respect to air velocities during case 1.

**Figure 5 foods-13-00901-f005:**
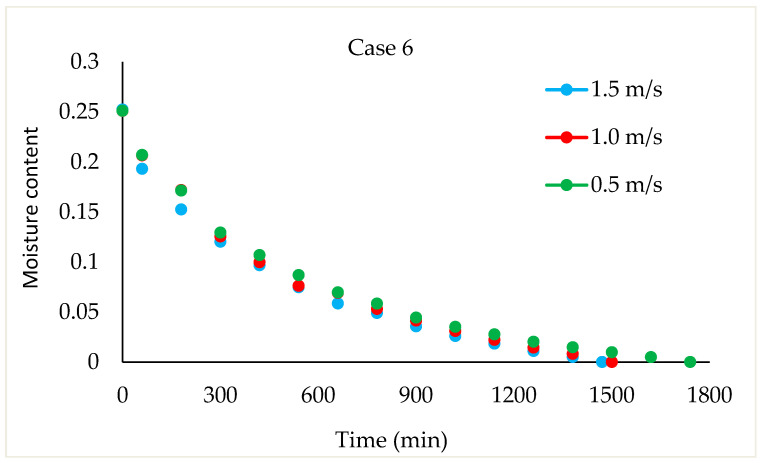
Change in time-dependent moisture content with respect to air velocities during case 6.

**Figure 6 foods-13-00901-f006:**
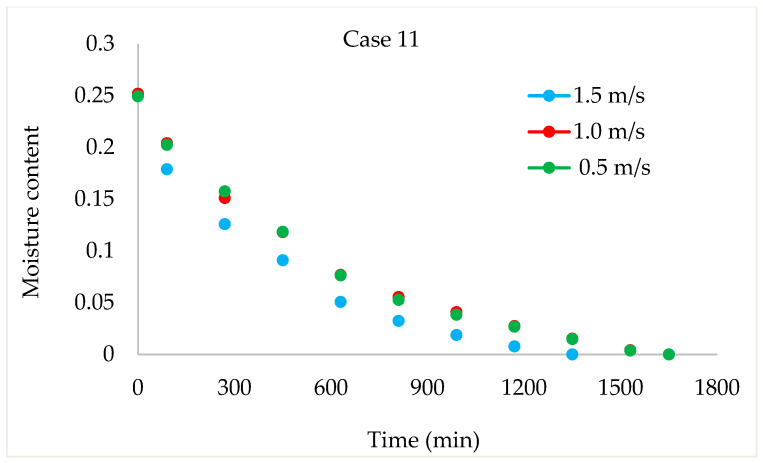
Change in time-dependent moisture content with respect to air velocities during case 11.

**Figure 7 foods-13-00901-f007:**
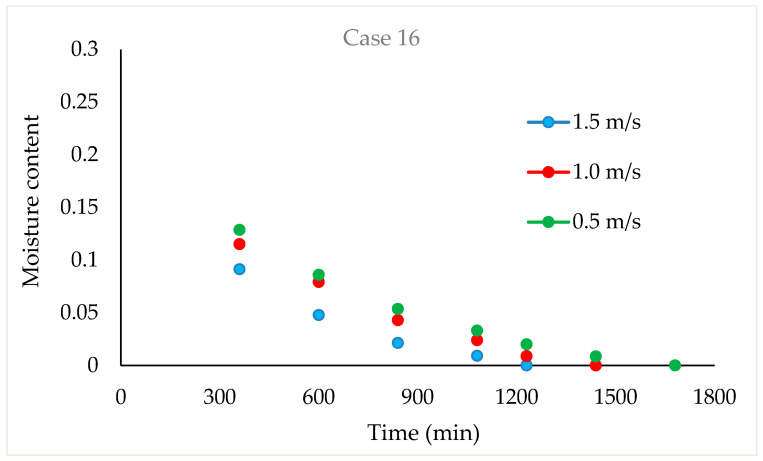
Change in time-dependent moisture content with respect to air velocities during case 16.

**Figure 8 foods-13-00901-f008:**
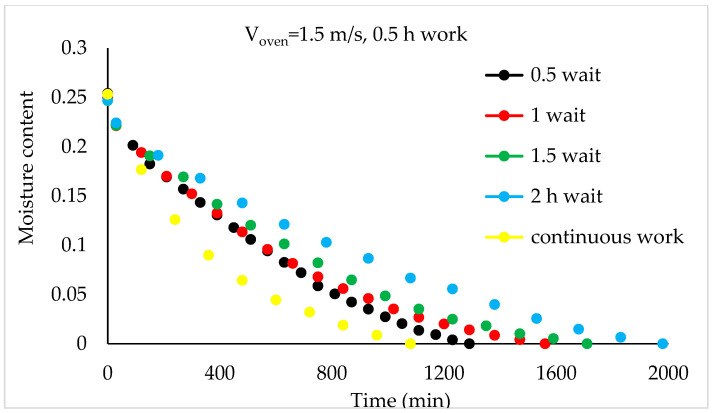
Variation in time-dependent moisture content of hazelnuts according to waiting time (0.5 h work).

**Figure 9 foods-13-00901-f009:**
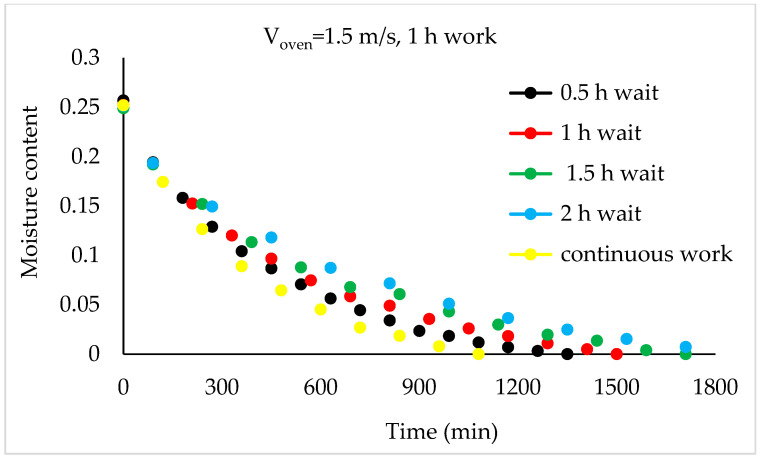
Variation in time-dependent moisture content of hazelnut according to waiting time (1 h work).

**Figure 10 foods-13-00901-f010:**
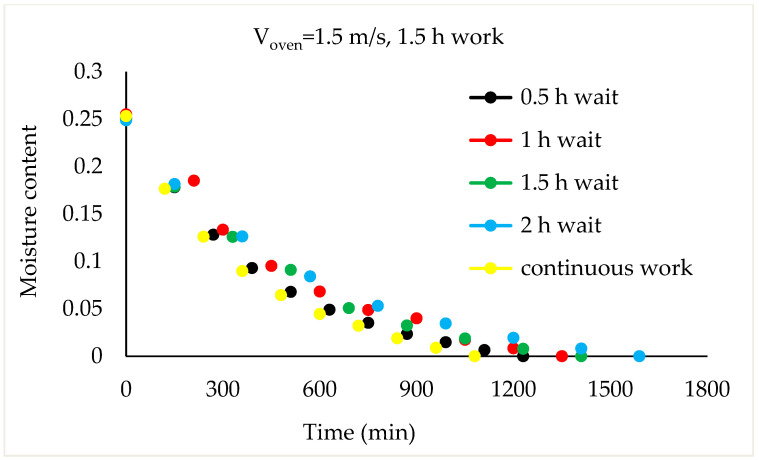
Variation in time-dependent moisture content of hazelnut according to waiting time (1.5 h work).

**Figure 11 foods-13-00901-f011:**
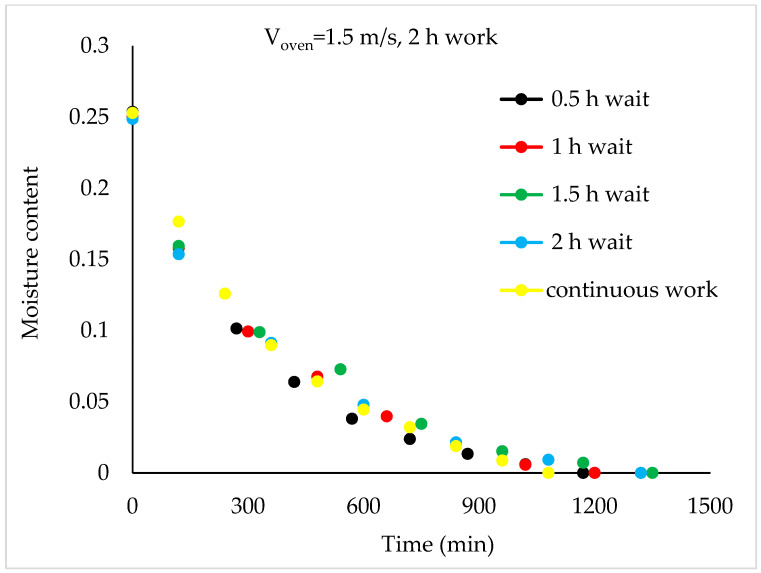
Variation in time-dependent moisture content of hazelnut according to waiting times (2 h work).

**Figure 12 foods-13-00901-f012:**
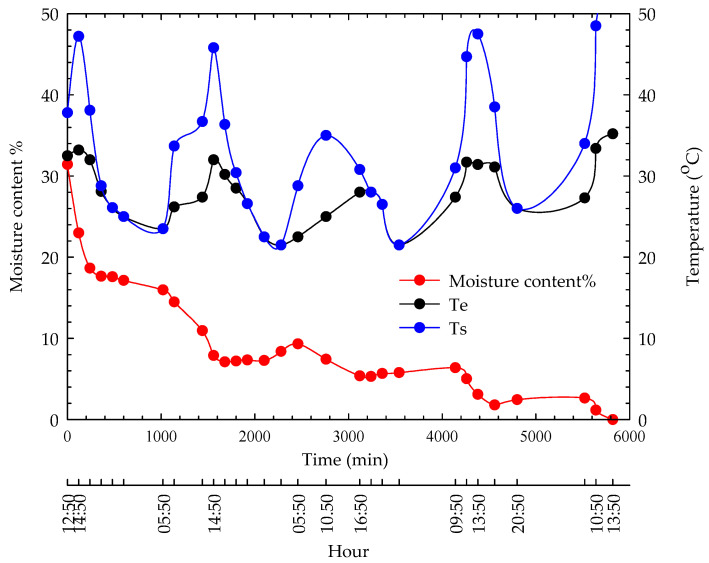
Time-dependent moisture content and temperature variations during sun-drying.

**Figure 13 foods-13-00901-f013:**
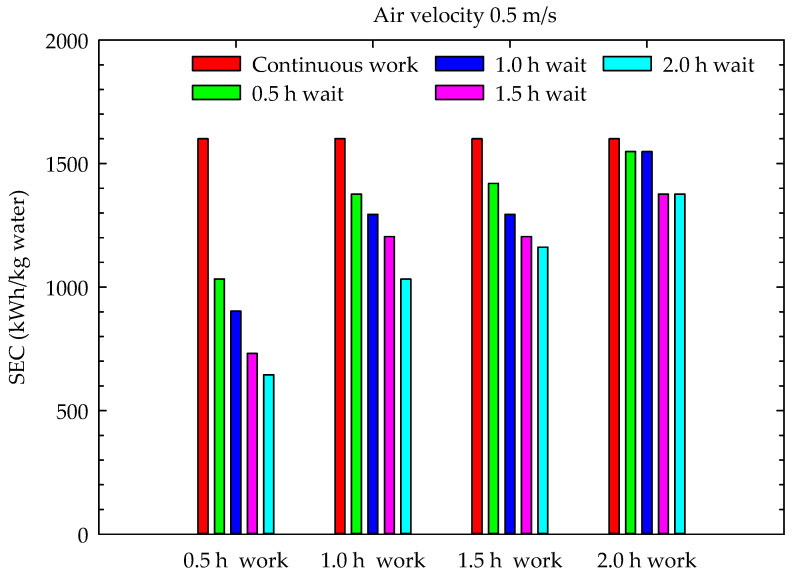
Specific energy consumption during drying (0.5 m/s air velocity).

**Figure 14 foods-13-00901-f014:**
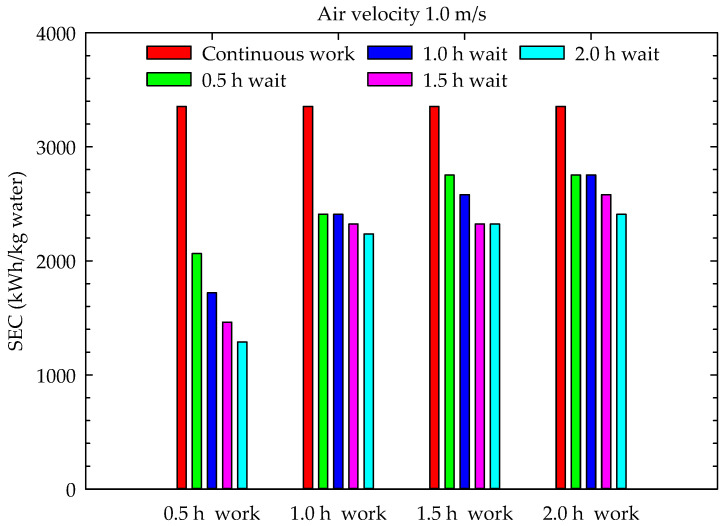
Specific energy consumption during drying (1 m/s air velocity).

**Figure 15 foods-13-00901-f015:**
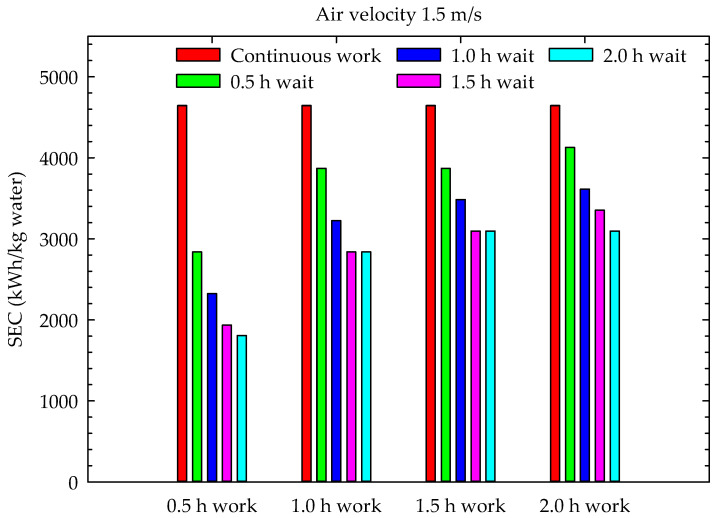
Specific energy consumption during drying (1.5 m/s air velocity).

**Table 1 foods-13-00901-t001:** Periodic drying experiment template.

Oven Working Time (Work) (h)	Idle Waiting Time (Wait) (h)			
	0.5	1.0	1.5	2.0
0.5	case 1	case 2	case 3	case 4
1.0	case 5	case 6	case 7	case 8
1.5	case 9	case 10	case 11	case 12
2.0	case 13	case 14	case 15	case 16

**Table 2 foods-13-00901-t002:** According to the drying periods, the working time of the oven, and the idle waiting time of the hazelnut outside the oven.

Drying Period	Vair = 0.5 m/s		Vair = 1 m/s		Vair = 1.5 m/s	
	Work (h)	Wait (h)	Work (h)	Wait (h)	Work (h)	Wait (h)
Case 1	720	690	720	570	660	630
Case 2	630	1200	600	1140	540	1020
Case 3	510	1560	510	1440	450	1260
Case 4	450	1750	450	1680	420	1560
Case 5	960	450	840	390	900	480
Case 6	900	840	840	780	750	630
Case 7	840	1170	810	1170	660	900
Case 8	720	1320	780	1260	660	120
Case 9	990	300	960	300	900	300
Case 10	900	540	900	540	810	600
Case 11	840	810	810	720	720	810
Case 12	810	960	810	960	720	960
Case 13	1080	240	960	270	930	210
Case 14	1080	480	960	420	840	360
Case 15	960	630	900	630	780	540
Case 16	960	810	840	720	720	600
Continuous work	1200	0	1170	0	1080	0

**Table 3 foods-13-00901-t003:** Energy utilization of periodic drying compared to continuous drying (0.5 m/s air velocity).

V_oven_ = 0.5 m/s	%ɳ_e_			
	0.5 h Wait	1.0 h Wait	1.5 h Wait	2.0 h Wait
0.5 h work	35.5	43.6	54.3	59.7
1.0 h work	14	19.1	24.8	35.5
1.5 h work	11.3	19.1	24.8	27.4
2.0 h work	3.3	3.3	14	14

**Table 4 foods-13-00901-t004:** Energy utilization of periodic drying compared to continuous drying (1 m/s air velocity).

V_oven_ = 1 m/s	%ɳe			
	0.5 h Wait	1.0 h Wait	1.5 h Wait	2.0 h Wait
0.5 h work	38.5	48.7	56.4	61.5
1.0 h work	28.2	28.2	30.8	33.3
1.5 h work	17.9	23.1	30.8	30.8
2.0 h work	17.9	17.9	23.1	28.2

**Table 5 foods-13-00901-t005:** Energy utilization of periodic drying compared to continuous drying (1.5 m/s air velocity).

V_oven_ = 1.5 m/s	%ɳ_e_			
	0.5 h Wait	1.0 h Wait	1.5 h Wait	2.0 h Wait
0.5 h work	38.9	50	58.3	61.1
1.0 h work	16.7	30.6	38.9	38.9
1.5 h work	16.7	25	33.3	33.3
2.0 h work	11.1	22.2	27.8	33.3

**Table 6 foods-13-00901-t006:** Oil and protein ratios of hazelnuts depending on drying periods (air velocity 1.5 m/s).

Drying Periods	%Oil Ratio	%Protein
Case 1	65.72	15.1
Case 2	64.68	15.8
Case 3	64.70	1.5
Case 4	65.98	16.6
Case 5	67.71	16.2
Case 6	64.80	17.3
Case 7	63.02	17.8
Case 8	65.84	16.1
Case 9	65.18	16.0
Case 10	61.46	15.4
Case 11	67.25	15.9
Case 12	64.39	16.7
Case 13	65.49	15.8
Case 14	63.92	16.7
Case 15	62.89	15.2
Case 16	66.93	14.8
Continuous work	65.38	14.2
Sun drying	62.48	17.2

## Data Availability

The original contributions presented in the study are included in the article, further inquiries can be directed to the corresponding author.

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
