# Peer review of "The Effect of the Periodic Drying Method on the Drying Time of Hazelnuts and Energy Utilization"

_foods, 2024, doi:10.3390/foods13060901_

Round 1

Reviewer 1 Report

Comments and Suggestions for Authors

In general, this manuscript presented relevant results about drying method of Hazelnuts.

In my opinion, this work is well written, but it is important to consider some suggestions, listed below.

Title: Lines 2-3

The drying behavior was not studied in this work, but the drying time, so I suggest change the title for: “The Effect of Periodic Drying Method on the Drying Time of Hazelnuts and Energy Utilization”.

Abstract: Line 9

It is necessary to add the objective of this work as: In this study, the effects of periodic drying of hazelnuts on their energy recovery, oil and protein content, as well as mass losses were carried out.

Introduction: Lines 96-97

I suggest changing this objective, as mass transfer was not studied here, but mass losses. The same observation for the food properties of hazelnuts, since there is no research into the food properties, but only the oil and protein content.

Material and Methods: Lines 120-122

I suggest including experimental information about the sun drying process, as well as temperature and velocity control.

Results and Discussion

It is necessary to first refer to the results obtained in the figures before presenting them in the text. Authors add figures before referencing them in the text. Change this for all figures in the text.

When studying drying processes, it is important to present the results as moisture content versus time and not mass loss versus time. Therefore, I suggest that the authors change the Figures 5 to 13, so that they can show the loss of moisture and not just mass loss.

Figure 13: Please clarify “Te” and “Ts” in footnote of this Figure.

It is not clear the difference between idle waiting time from Table 1 and Table 2. Please clarify. Besides, the authors inform that (lines 131-132) “The idle waiting process was carried out at ambient temperature (approximately 30oC) outside the oven with the hazelnut covered”. It is important to explain how this process was carried out. The hazelnut samples were completely removed from the dryer, covered, and added to which location?

Table 3. The correct is 1.5 h work, not 15h, please change it.

It is important to explain in the text the relevance to determine oil and protein ratios of hazelnuts. I suggest including the hazelnuts composition in this work.

Author Response

The attachment to the file includes not only the answers given to the relevant reviewer but also the answers given to all reviewers.

Reviewer 2 Report

Comments and Suggestions for Authors

The Effect of Periodic Drying Method on the Drying Behavior of Hazelnuts and Energy Utilization (Manuscript number – 2895384) is the study about drying of hazelnuts using periodic drying (16 drying protocols) at three different air velocities and comparing the results with continuous drying and solar drying to see the effect on drying characteristics, drying time, energy consumption, oil and protein content of the final product. This is an interesting study to see the use of periodic drying protocols to save energy consumption and product quality. However, following points were noted while reviewing the manuscript, which need amendment before accepting for publication in Foods:

Introduction: At present, it contains 9 paragraphs; this section should be 2-4 paragraphs (maximum). Please re-write this section to reduce the number of paragraphs by merging these paragraphs into 2 to 3 or maximum 4 paragraphs. Also, please clearly indicate the purpose of this study to fulfil the existing knowledge gap in hazelnut drying.

Section 2.2: please cite the reference for equation 1 to 7.

At the end of Materials and Methods section, please add Statistical Analysis section.

Section 3.1: The remaining results for case 2, 3, 4, 5, 7, 8, 9, 10, 12, 13, 14 and 15 can be presented as a supplementary file.

Table 3, 4, 5, and 6: Results should be presented with “mean value ± standard deviation” and the mean values should be tested for significance (P < 0.05 or P > 0.05). All bar diagrams should be presented with error bar with significance test results.

Conclusion: Please write this section in a paragraph; instead of point-wise.

English language and grammar need thorough revision for minor mistakes and typos; for example, oC Vs. C Vs. oC; properties Vs. proper-tise (line 54); hazelnut Vs. hazel-nut (line 79); Solar drying Vs. Drying sun (Table 6); use of present tense for presenting the observed results etc.

Comments on the Quality of English Language

Moderate language polishing is recommended. 

Author Response

(The authors gave the same response as above.)

Reviewer 3 Report

Comments and Suggestions for Authors

Prior to publication, some necessary changes and corrections have to be made:

- fig.3 is not necessary and not referenced in text.

- symbols in equations have to be described.

- eq. 1:  E is not the energy but the power (energy flow rate)

- eq. 2: if V should be the velocity, use v instead

constants below eq. 2: what about taking the air humidity into account?

- eqs. 5 and 6: what should be tc and tp?

-ll.166: figs. 3 and 4 aren' t the right references.

- all figs. with drying curves:
instead of the sample mass, the water content (d.b. or w.b.) should be uses.  

ll. 214: what is the air humidity?

fig. 14: why is the energy flow divided by the area is used? The energy flow is senseless, here. Use the energy, instead. For comparabilty the energy used for the evaporation of 1kg water should be used.

What about the energy consumption of fans ...?

chapter 3.3:

why should the protein and oil content be dependent on drying? Isn't it just the different water content or different samples?

Author Response

(The authors gave the same response as above.)

Round 2

Reviewer 2 Report

Comments and Suggestions for Authors

Title: The Effect of Periodic Drying Method on the Drying Behavior of Hazelnuts and Energy Utilization

Manuscript number: 2895384

After assessing the revised version of above manuscript, I found that the authors addressed all the issues raised satisfactorily, except the one concerning Table 3, 4 and 5.

The authors were asked to present the data as “mean ± standard deviation” for an individual set of data (not to calculate the “mean ± standard deviation” for all sets in combinations!). For example, the authors were asked to calculate “mean ± standard deviation” for three data for “0.5 h work and 0.5 wait” set. Similarly, the authors were asked to calculate “mean ± standard deviation” for three data for “0.5 h work and 1.0 h wait” and so on…… To calculate “mean ± standard deviation” at once for all sets of data (horizontally and vertically) has no meaning. Please do needful action accordingly. Once this problem is sorted out, I recommend this manuscript to accept for publication in Foods.

Author Response

(The authors gave the same response as above.)

Reviewer 3 Report

Comments and Suggestions for Authors

The paper has already improved very much by the revision. Nevertheless, a comparability of the energy used for evaporation a certain amount of water out of the hazelnuts with other studies is still missing.

Furthermore it still stays unclear, why the energy used is given relative to a certain drying area, here. Does the area change during the experiments?

Concerning the protein and fat contents, the variability of the raw product would be helpful to see, if differences really stem from drying or just from differences in the raw material.

Author Response

(The authors gave the same response as above.)
